# Measurement and Correction of Pointing Error Caused by Radio Telescope Alidade Deformation based on Biaxial Inclination Sensor

**DOI:** 10.3390/mi14071283

**Published:** 2023-06-22

**Authors:** Qian Xu, Fei Xue, Hui Wang, Letian Yi

**Affiliations:** 1Xinjiang Astronomical Observatory, Chinese Academy of Sciences, Urumqi 830011, China; 2Key Laboratory of Chinese Academy of Sciences for Radio Astronomy, Urumqi 830011, China; 3Xinjiang Key Laboratory of Radio Astrophysics, Urumqi 830011, China

**Keywords:** inclinometer, pointing errors, radio telescope, alidade, thermal gradients, track unevenness

## Abstract

One of the key reasons for the deterioration of antenna pointing accuracy for radio telescopes is the deformation and tilt of antenna alidades, which primarily result from track unevenness and thermal gradients. A high-precision inclinometer measurement system is installed to investigate the tilt of the antenna alidade and the pointing errors caused thermally. An environment control box with a leveling base was designed to reduce the interference of the external environment, which proved to be effective in guaranteeing the zero-point stability and repeat accuracy of the inclinometer. The tilt of the alidade caused by the track unevenness was measured by a test of slowly rotating the antenna along the azimuth at windless nighttime. A 5-day antenna stationary test and a 48 h astronomical pointing error measurement were performed, which proved the inclinometer measurement system is capable of measuring the thermally induced inclinations with acceptable accuracy. Through a preliminary compensation experiment, the pointing error is compensated from 37″ to 12″, which shows that the application of the system has a good effect on improving the pointing accuracy of the antenna. The system with high measurement accuracy, good system stability, and low computational complexity, proves an effective tool for the radio telescope to solve the problem of real-time measurement and compensation for antenna pointing errors.

## 1. Introduction

A radio telescope is a scientific instrumentation for observing the electromagnetic signals from the universe. As shown in Figure 1, most large-aperture radio telescopes, though not all, choose an alidade-supported antenna, which consists of an alidade, a reflector, an azimuth axis, an elevation axis, wheels on the track, and other components.

By driving azimuth and elevation mechanisms to rotate around the azimuth axis and the elevation axis separately, the antenna reflector can be pointed at the astronomical source to be observed. If there is a deviation between the commanded position of the antenna and the actual position of the astronomical source, then the radio telescope has an antenna pointing error, as shown in Figure 2. Pointing error has two components: elevation error and azimuth error (or cross-elevation error). In order to ensure the sensitivity and resolution of radio telescopes, it is desirable that the antenna pointing error should not exceed 1/10 of the beam width (half power beam width, HPBW) [1], which brings strict requirements for radio signal observation above the centimeter wavelength band. For example, when operating at 100 GHz, a pointing accuracy of 1″ is reasonable.

From the perspective of antenna components, a major source of pointing errors is the antenna alidade [2,3,4]. The alidade is the foundation of the antenna, an important part of the antenna azimuth and elevation motion mechanism since it plays a significant role in ensuring perfect orthogonality between the azimuth axis and the elevation axis of the antenna. The pointing errors caused by the alidade are mainly caused by two reasons: one is the real-time change in the environmental load, such as temperature gradients [5,6,7,8], and the other is by the track unevenness [9,10,11]. When there are temperature gradients on the alidade due to solar irradiation, or unevenness on the track, the alidade will deform and tilt, resulting in non-orthogonality of the axis system, and further pointing error. Some experimental and simulation studies prove the pointing error caused by solar radiation and orbital unevenness account for a large part of the antenna’s overall pointing error.

One way to obtain the overall pointing errors of a radio telescope is to use astronomical measurement [12,13,14], such as the cross-scan method which measures the deviation between the center of the signal peak and the antenna instruction position by scanning the radio source in two orthogonal directions and performing a multi-point Gaussian fitting. However, performing an astronomical measurement needs to suspend the scheduled astronomical observation and arrange a specific period of time, which brings a loss of effective observing time. Moreover, astronomical measurement is sensitive, which makes the measurement results of one period not instructive for observation in other periods when the external environmental conditions change. For the above two reasons, it is impossible to achieve real-time pointing error measurement without interrupting planned observation using astronomical methods. Therefore, it is necessary to explore a pointing error measurement method that can be carried out under all weather conditions and parallel to scheduled astronomical observations.

The biaxial inclinometer is a high-precision inclination sensor that simultaneously measures the small inclinations of its x- and y-axis. The horizontal plane of an inclinometer is a fluid surface that remains horizontal irrespective of the inclination of the sensor so that the angle of inclination corresponds to the angle between the surface of the fluid and the base of the sensor. Some commercial inclinometers can reach a resolution of 0.001 mrad, or 0.2 arcseconds [15]. The ability of an inclinometer to measure small inclinations makes it suitable for tilt measurement of antenna alidade, by which the pointing error resulting from the tilt of the alidade can be inferred. A series of studies have been carried out to measure the tilt of the alidade using an inclinometer and indirectly infer the antenna pointing errors [16,17,18,19,20,21]. Still, the accuracy of the data obtained from these experiments is not sufficient for pointing error correction. One possible reason is that the unstable thermal environment compromises the zero-point stability of inclinometers. One study has shown that an inclinometer can be calibrated with empirical formulas using data measured over several nights [22], but there is still a lack of evidence that the calibrated inclinometer is capable of maintaining stability when experiencing seasonal temperature variation.

Our goal is to take full advantage of the high precision of the inclination sensor. An inclinometer measurement system with an environmental control box (ECB) is designed and tested based on the 25 m Nanshan Radio Telescope (NSRT). The NSRT (as shown in Figure 3) is located at the Nanshan Observatory in Urumqi, Xinjiang, China, and can operate at frequencies up to 43 GHz [23,24]. At present, the average pointing error of NSRT is greater than 15″. A stationary test was performed, which proves that the zero-point stability and repeatable accuracy with the application of the ECB are at the same level as those measured in an indoor environment. Multiple tests were conducted to demonstrate that the inclinometer measurement system is able to infer the pointing errors caused by the tilt of the alidade due to track unevenness and thermal gradients with satisfactory accuracy.

## 2. Measuring System

### 2.1. Environmental Control Box

The high-precision biaxial inclinometer’s absolute accuracy is significantly affected by the level of its installation, as well as the ambient temperature and humidity during measurement. For this experiment, we utilized the LEICA Nivel-210, a commercially available two-axis precision inclination sensor with a resolution of 0.001 mrad. The main technical parameters of the sensor are listed in Table 1. To maintain optimal measurement accuracy, a flatness tolerance within 0.01 mm and a platform inclination of ±1.51 mrad are required. The sensor’s operational temperature range is −20 °C to 50 °C, with a zero-point error of 0.005 mrad induced by every 1 °C of temperature drift. The seasonal temperature range at Nanshan Observatory, located in Urumqi, China, spans from −40 °C to 60 °C. The significant disparity in temperature can have a notable impact on the precision of inclinometer measurements and may even result in irreparable harm to its internal optical components. Therefore, it is imperative to implement stringent measures for the environmental protection of the inclinometer. An ECB was designed and constructed to ensure optimal functioning and precise performance of the inclinometer. The components of the ECB are shown in Figure 4a, which includes a base (1), a seal ring groove (2), three sets of leveling screws (3), temperature and humidity sensors (4), data acquisition and temperature control PCB (5), a relay switch (6), four sets of heating modules (7), an inner protective cover (8), and an outer protective cover (9).

The ECB is depicted in Figure 4b post-manufacturing and assembly. The shell of the ECB has extremely high self-weight and stiffness to avoid interference with sensor measurements. The inclinometer’s level is achieved by adjusting the three sets of leveling screws, ensuring that it falls within the expected range for accurate measurements, and keeping stability. A temperature and humidity sensor, along with a microcontroller unit (MCU), is integrated into a printed circuit board (PCB) within the enclosure. The outer protective cover is equipped with two waterproof interfaces, one of which provides a 12 V/DC linear DC power supply for control and communication, especially for high-precision sensor power supply, while the other supplies 220 V/AC power to four heating modules located on the side of the inner protective cover. The inner protective cover not only protects the inclinometer but also plays a role in evenly conducting heat and insulation. As depicted in Figure 4c, the interior of the outer protective cover and base plate of the inclinometer has been coated with a layer of white aerogel blanket, which serves as another effective means of passive temperature control.

The communication protocol and wiring path of the measurement system are illustrated in Figure 5. A personal computer is connected to the MCU via a TCP/IP to RS422 bus converter, which then connects to the inclinometer through an MCU-encoded RS422 to RS232 converter, facilitating data transmission between the computer and inclinometer. The temperature and humidity inside the box, measured by the sensor on the PCB, are transmitted to the MCU in real time. Once the MCU acquires temperature data from the sensor, it will compare it with the pre-set temperature threshold. The MCU also includes a logic program to control the activation and deactivation of relays for heating or stop relays as a means of active temperature control. In this experiment, the heating temperature threshold was established at 30 °C. When the temperature falls below 30 °C, the MCU maintains relay closure to activate heating modules. Conversely, the relay is deactivated, and heating modules are shut down to ensure that the internal environment of the inclinometer remains close to 30 °C.

The schematic diagram in Figure 6 illustrates the user interface of the acquisition software, which enables real-time display and storage of data from inclinometers as well as temperature/humidity sensors. Additionally, a configuration interface facilitates customization of IP address, port number, sampling rate, and heating temperature threshold.

### 2.2. Error Analysis of Install Angle

The desired mounting angle of the inclinometer is shown in Figure 7, where the x-axis of the inclinometer is aligned with the elevation axis of the antenna.

According to reference [10], the azimuth pointing error Δ_az_ and elevation pointing error Δ_el_ of the inclinometer installed in this orientation at a given elevation angle θ_el_ can be expressed as follows:Δ_el_ = α_x_,(1)
Δ_az_ = −α_y_·tanθ_el_(2)

The values of α_x_ and α_y_ represent the tilt angle of the measuring plane around the inclinometer’s x-axis and y-axis, respectively. However, it is hard to achieve perfect parallelism between the x-axis and the elevation axis. The subsequent analysis shows the extent of measurement error resulting from a minor deviation in installation angle.

As illustrated in Figure 8, OAB denotes the installation plane of the inclinometer. While OA represents the optimal installation direction (i.e., parallel to the elevation axis of the antenna), the x-axis of the inclinometer is aligned with the OB direction and OC is perpendicular to OB, paralleling with the y-axis of the inclinometer. When the plane tilts around the OA, point B rotates to point B’, and point C rotates to point C’. The inclinometer output represents the rotation value around the OB axis, known as ∠COC’. However, our objective is to determine the angle of rotation, ∠B’AB, with respect to the direction of the pitch axis.

Set ∠B’AB = α, ∠AOB = β, ∠COC’ = γ, |OA| = l, |AB| = m.

Obviously |OB’| = |OB| = $\sqrt{l^{2}+m^{2}}$, and ∠AOB’ = ∠AOB = β

So |OC| = |OC’| = |OB’|·tan∠AOB’ = |OB’|·tan∠AOB = |OB|·tanβ.

Since the distance from point C and point B to the axis of rotation OA are equal (both being m), then: |CC’| = |BB’| = $2\left|AB\right|sin\frac{∠B’AB}{2}$= $2m\cdot sin\frac{\alpha }{2}$
$$sin\frac{∠C’OC}{2}=sin\frac{\gamma }{2}=\frac{1}{2}\cdot \frac{\left|CC’\right|}{\left|OC\right|}=\frac{1}{2}\cdot \frac{2m\cdot sin\frac{\alpha }{2}}{\left|OB’\right|tan\beta }=\frac{m\cdot sin\frac{\alpha }{2}}{\sqrt{l^{2}+m^{2}}tan\beta }$$

Since $\frac{m}{\sqrt{l^{2}+m^{2}}}$= sinβ, so
$$sin\frac{\gamma }{2}=\frac{sin\beta sin\frac{\alpha }{2}}{tan\beta }=sin\frac{\alpha }{2}cos\beta$$

Due to the small value of the measured angle α, it is reasonable to approximate $sin\frac{\alpha }{2}\approx \frac{\alpha }{2}$, $sin\frac{\gamma }{2}\approx \frac{\gamma }{2}$. therefore, we obtain
γ = αcosβ

The absolute value of relative error ε is:$$\varepsilon =\left|\frac{\alpha -\gamma }{\alpha }\right|=\left|\frac{\alpha -\alpha cos\beta }{\alpha }\right|=\left|1-cos\beta \right|$$

It can be inferred from the formula of ε that the relative measurement error is merely 1.5% when the installation error angle is 10° and 6.1% when it is 20°. Given that the range of the sensor is only ±1.51 mrad, it may be deemed negligible in terms of the inclinometer’s installation-induced measurement errors.

### 2.3. Measurement System Stability Test

A stability test was conducted to calibrate and validate the zero-point stability of the measurement system. The antenna remained stationary for three consecutive windless nights, with a fixed azimuth angle of 70° and elevation angle of 90°. Ambient temperature data from these three nights were recorded, as depicted in Figure 9.

The external temperature conditions remained highly consistent over the course of three consecutive nights. To verify system stability, we conducted the following tests: on the first and third nights, the temperature control switch was activated, while on the second night, it was deactivated. The inclinometer data were recorded and are presented in Figure 10.

It is evident that the zero drift of the inclinometer on the second night, when compared to that of the first and third nights, becomes more pronounced upon turning off the temperature control switch. This observation confirms that external environmental factors do indeed impact the stability of the inclinometer zero point. The root mean square (RMS) values of the inclinometer measurements taken during the three nights are presented in Table 2. The zero-point RMS value of the inclinometer on the first and third nights is significantly lower than that of the second night. It is evident that activation of the temperature control module significantly mitigates zero-point drift in the inclinometer, thereby affirming that the environmental control box ensures comparable repeatability for outdoor measurements as those conducted in laboratory settings.

## 3. Tilt and Deformation of Alidade Caused by Track Unevenness

Experimental measurements were conducted to quantify the tilt and deformation of the alidade induced by track unevenness, while the antenna was rotating in azimuth (AZ) at a constant speed of 0.1 °/s from AZ 0° to AZ 360°. The inclinometer’s sampling rate was set to 1 Hz, and the test was conducted on a windless night to eliminate temperature gradient interference. To verify the measurement system’s repeatability, the experiment was performed three times in succession, yielding results as shown in Figure 11.

Figure 11 depicts the unfiltered raw data, which exhibits excellent repeatability in tilt meter measurements. This further validates the effectiveness of the inclinometer environment control system in mitigating external temperature drift-induced noise. The peaks observed at 0° and 360° are attributed to the antenna moment of inertia when starting and stopping, rather than noise. Table 3 presents the RMS values of antenna pointing errors in both elevation (EL) and azimuth (AZ) directions, which are derived from inclinometer measurement data using Equations (1) and (2).

The table reveals that the track unevenness exerts an 11.3″ impact on the antenna’s EL direction and an 11.8″ effect on its AZ direction. It is worth noting that, in actual astronomical observations, some of the pointing errors caused by orbit have been compensated by the pointing model.

## 4. Pointing Errors Caused by Thermal Gradient

On a clear day, solar radiation induces thermal gradients on the alidade, leading to non-orthogonality between the EL and AZ axes and deterioration of pointing accuracy. Consequently, astronomical observations at NSRT higher than K and Q bands requiring greater pointing accuracy can only be conducted at night. To measure pointing errors caused by thermal gradients, we designed a measurement system and carried out experiments.

### 4.1. Pointing Errors When Antenna Is Stationary

An experiment was conducted to assess the thermal deformation caused by solar radiation when the antenna remained stationary for five consecutive clear and windless days, from 4 February to 8 February 2019. The AZ and EL angles of the antenna were maintained at 66° and 88°, respectively. Meanwhile, the measurement system continuously recorded data at a sampling rate of 1 Hz. Figure 12 displays satisfactory unfiltered data.

As depicted in Figure 12, the antenna alidade exhibits a relatively stable behavior during the non-sunlight period from 22:00 to 10:00. However, it experiences significant tilting between 10:00 to 20:00 due to the non-uniform temperature distribution caused by solar radiation. On average, the five-day experimental data showed an increase in AZ direction tilting at approximately 10:00, peaking at 53.4″ around 14:00, and gradually stabilizing after 18:00. From 10:00 to 18:00, an average azimuth error of 35.7″ was observed. In terms of EL direction tilting, it began at 12:00 and peaked at around 44.5″ by about 16:00 before becoming steady around 20:00. The average tilt in EL direction is 24.4″ between 12:00 and 20:00. It is evident that the onset and peak of EL tilting occur approximately two hours later than those of azimuthal error, which can be attributed to the relationship between antenna alidade orientation and apparent solar motion.

Taking February 4th as an example, Figure 13 illustrates the orientation of the antenna alidade and the apparent motion curve of the sun. The antenna remained stationary while facing northeast at a bearing of 66°. After 09:23, the sun initially illuminates the S-E side of the alidade, resulting in a temperature differential between the S-E and N-W sides that induces rotation around the AZ axis. As a result, the AZ tilt value rapidly increases after sunrise and reaches its zenith at noon (around 14:23). As the sun gradually shifts towards the S-E after noon, the temperature on that side increases compared to the N-E, resulting in a peak tilt in the EL direction when directly radiated by sunlight from the S-W side. This phenomenon explains why there is approximately a two-hour delay between EL tilting and AZ tilting. It should be reminded that the noon time at Urumqi Nanshan Observatory is about 14:00 (UTC+8). This is due to the fact that Urumqi Nanshan Observatory is located in the UTC+6 time zone, while measurement data are recorded using Beijing time, which corresponds to UTC+8.

### 4.2. Inclinometer Measurement Compared with Astronomical Observation

To verify the applicability of the inclinometer measurement system in measuring pointing errors during astronomical observation, an experiment was conducted from 12:00 on 14 February 2019, to 12:00 on 16 February 2019. The pulsar 1800 + 7828, located near the North Pole, was subjected to continuous K-band observation for a period of 48 h during which pointing error was measured using the cross-scan method. Concurrently, the inclinometer measurement system was activated to sample the alidade tilt at a rate of 1 Hz. The Pulsar 1800 + 7828 is highly suitable for this experiment due to its strong flux in K-band. Its proximity to the North Pole on the celestial sphere results in minimal changes in antenna azimuth and elevation during observation, as depicted in Figure 14. The antenna’s azimuth shifts by only ±15°, rendering any pointing errors caused by track unevenness negligible.

As the weather remained sunny and windless throughout the 48 h experiment, it can be inferred that thermal gradients were primarily responsible for the antenna pointing errors observed. The inclinometer measurement data were filtered and compared with astronomical observations to generate the curve depicted in Figure 15.

The inclinometer measurement system demonstrates a high level of agreement with astronomical observation measurements in terms of pointing errors, as illustrated by Figure 15. Even during the period from 12:00 to 18:00 when thermal gradients significantly impact the antenna alidade, the difference between pointing errors obtained through the measurement system and those derived from astronomical observation is only 2″ (RMS). It illustrates that the experiment exhibits thermally induced pointing errors, which peak earlier in the AZ direction than in the EL direction. This phenomenon is similar to that observed in Section 4.1 when the alidade was held stationary. Figure 16 depicts a schematic diagram illustrating the correlation between the apparent solar motion and the orientation of the alidade on 14 February. The rapid increase in antenna pointing errors after 9:00 is attributed to the sun’s illumination of the N-E side of the alidade post-sunrise, causing an elevation in temperature that surpasses other areas, thereby resulting in an initial rise in azimuth direction for pointing errors. As the sun gradually shifted to illuminate the southeast side, pointing errors in elevation direction increased rapidly. As it was noon at this time, the thermal gradients on the alidade intensified, resulting in greater pointing errors in elevation than azimuth direction.

The difference between pointing errors obtained from inclinometer measurement and those measured by astronomical observation during the period of 12:00 to 18:00 may be attributed to thermal gradients causing deformation in other parts of the antenna, particularly changes in the position of the secondary reflector or shape of the main reflector. The experiment clearly demonstrates that the thermal deformation of the alidade is a significant factor affecting the daytime pointing accuracy of the antenna. As this aspect of pointing deviation can be gauged by an inclinometer, it can be rectified using data from the said instrument.

As the stability of the inclinometer measurement system has been demonstrated in previous experiments and its accuracy has been verified through astronomical observations, it shows great potential for correcting thermally induced pointing errors during daytime. The same observation was conducted over the following two days, with the antenna pointing errors pre-compensated based on results from the experiment on 14 February. The pointing error comparison between the two astronomical observations is illustrated in Figure 17.

It is evident from Figure 17 that the pointing accuracy has significantly improved after compensation, particularly during the period from 14:00 to 17:00 when thermal gradients have the greatest impact on antenna performance. Table 4 presents a comparison of RMS values for pointing errors before and after compensation.

## 5. Conclusions

This paper presents a system for measuring pointing errors based on a biaxial inclinometer and an environment control box, which enables the measurement of alidade tilt caused by track unevenness and thermal gradients.

The analysis of the installation angle error for the inclinometer indicates that a slight deviation from the preset angle does not result in significant measurement errors. The implementation of an environment control box effectively reduces system noise and improves zero-point stability, enabling accurate data filtering without loss of real-time information, and has been proven through a stability test.

The inclinometer measurement system demonstrates the impact of track unevenness and thermal gradients on the antenna alidade, with its accuracy validated by data obtained from observation and pointing measurements of Pulsar 1800 + 7828. Compared to the method of calculating structural deformation to derive pointing errors through measuring temperature gradients by sensors, this method is more direct and accurate, with less data computation and faster speed, and can be easily deployed on the antenna. Through further improvements, this is the most likely way to achieve real-time online compensation for pointing errors. The implementation of this system can significantly enhance daytime antenna pointing precision.

## Figures and Tables

**Figure 1 micromachines-14-01283-f001:**
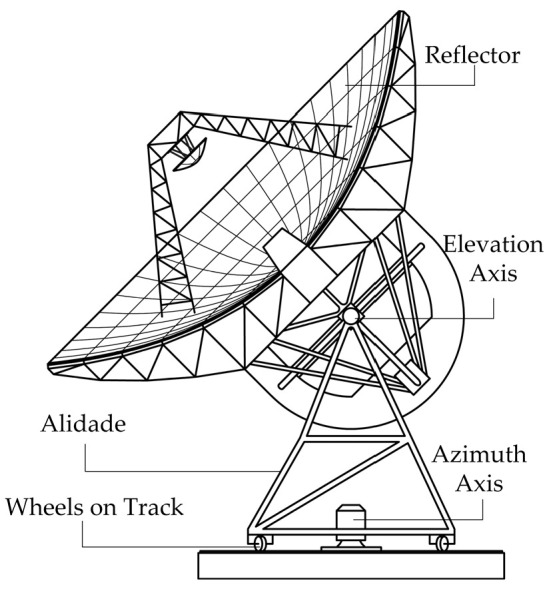
The components of a radio telescope.

**Figure 2 micromachines-14-01283-f002:**
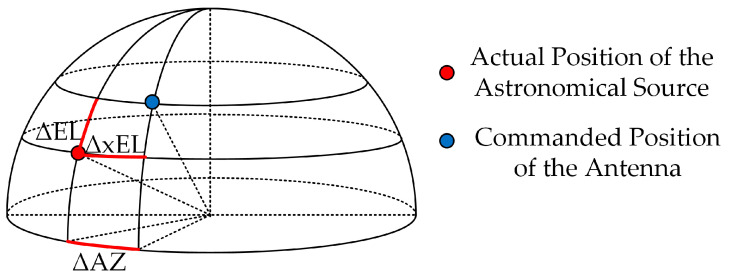
The antenna pointing errors of a radio telescope.

**Figure 3 micromachines-14-01283-f003:**
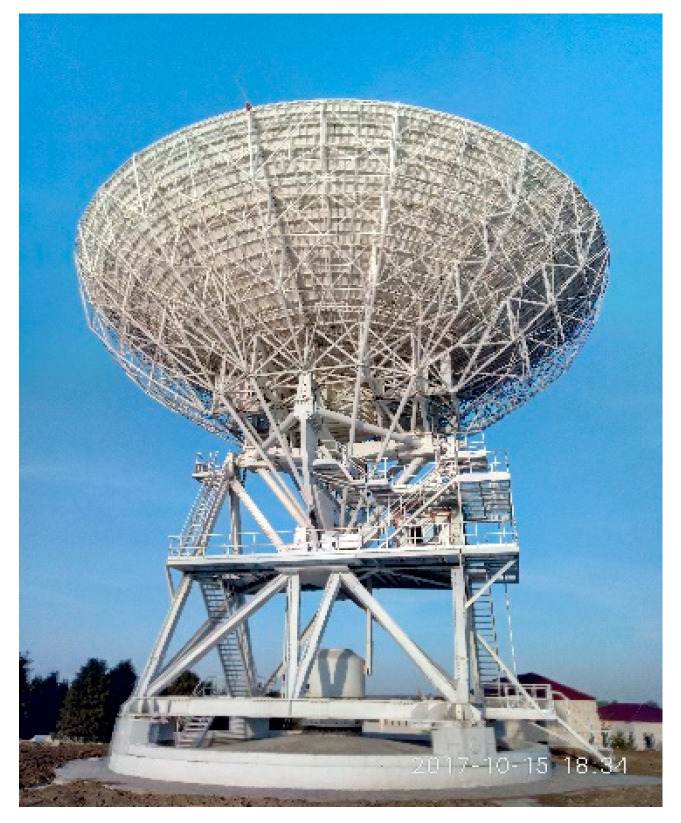
The 25 m aperture NSRT.

**Figure 4 micromachines-14-01283-f004:**
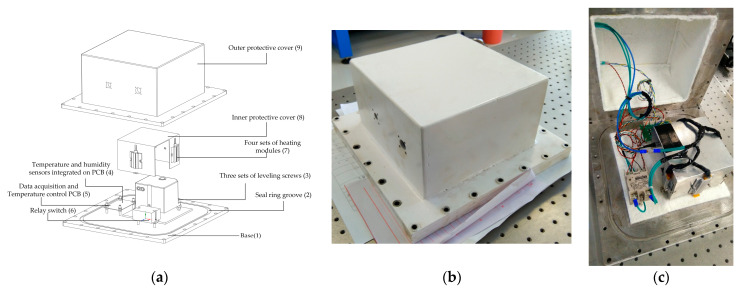
The environmental control box of the inclinometer: (**a**) the components of environment control box; (**b**) when the outer protective cover is closed; (**c**) when the outer protective cover is opened.

**Figure 5 micromachines-14-01283-f005:**
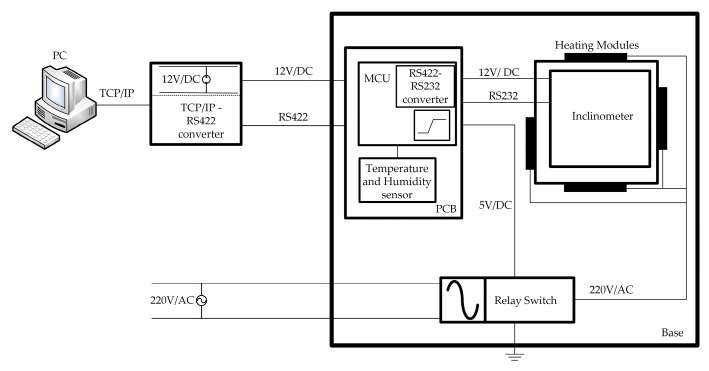
The communication protocol and wiring path of the measurement system.

**Figure 6 micromachines-14-01283-f006:**
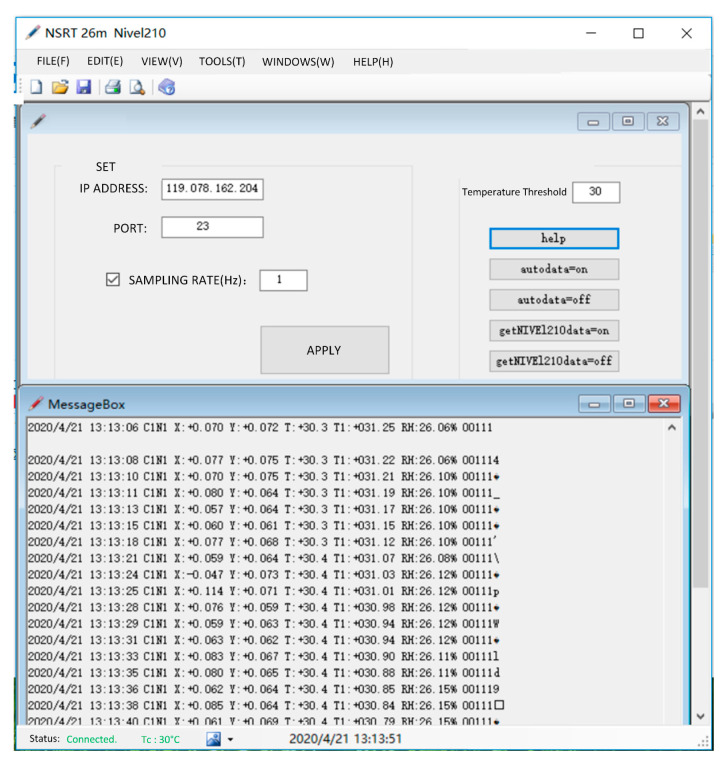
A schematic diagram of the user interface of the acquisition software.

**Figure 7 micromachines-14-01283-f007:**
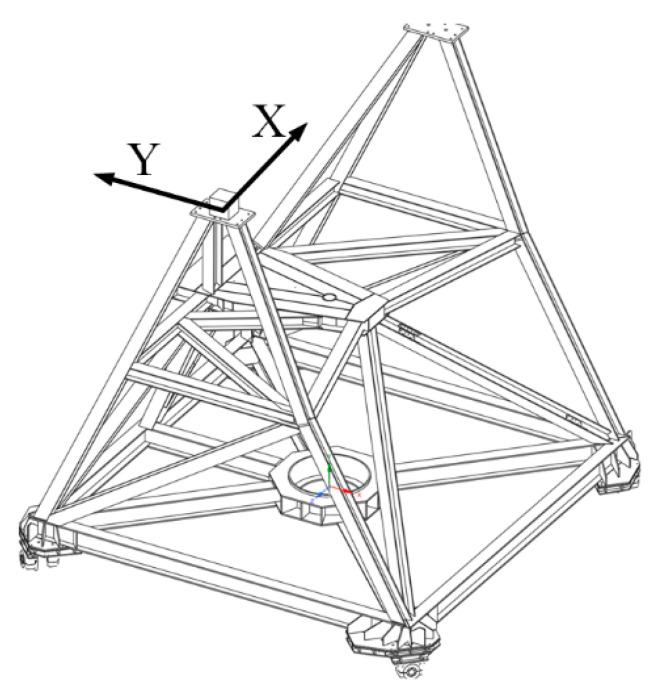
The ideal installation angle of the inclinometer.

**Figure 8 micromachines-14-01283-f008:**
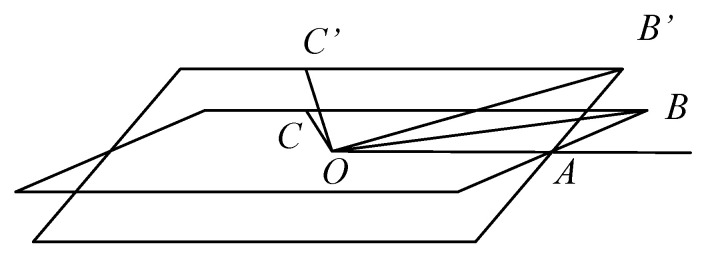
Schematic of error analysis.

**Figure 9 micromachines-14-01283-f009:**
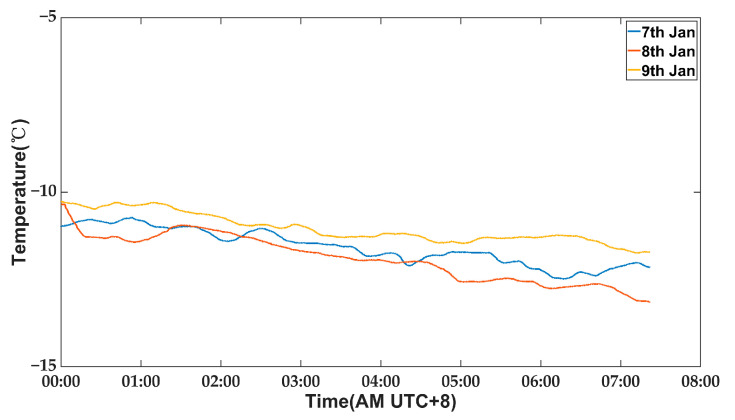
The ambient temperature data of the three nights.

**Figure 10 micromachines-14-01283-f010:**
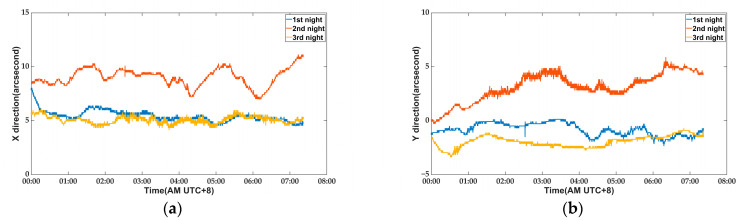
The data recorded by the inclinometer at three nights: (**a**) tilt in X direction; (**b**) tilt in Y direction.

**Figure 11 micromachines-14-01283-f011:**
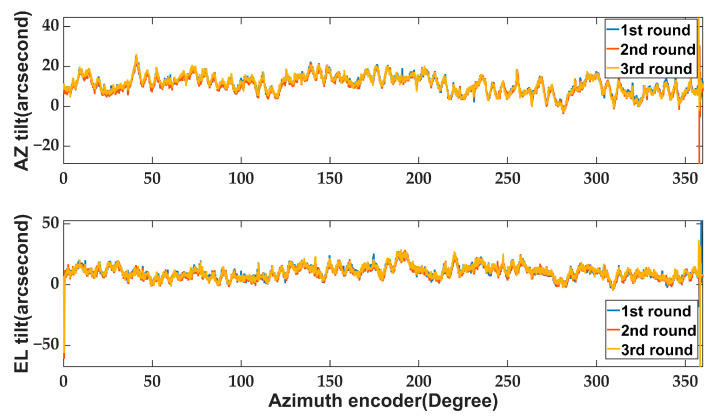
The measurement repeatability of three rounds: tilt in AZ and EL direction.

**Figure 12 micromachines-14-01283-f012:**
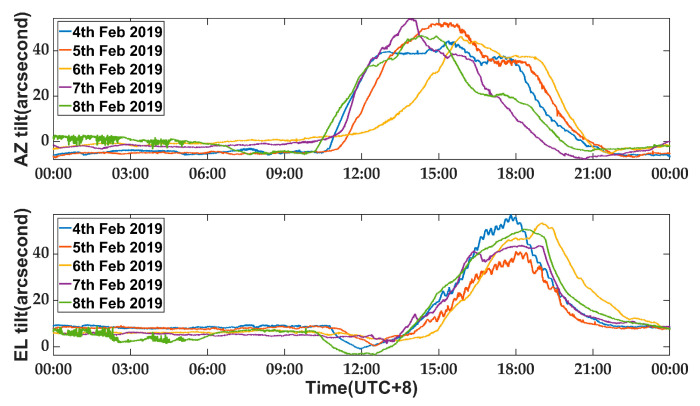
Thermal deformation caused by solar radiation: tilt in AZ and EL direction.

**Figure 13 micromachines-14-01283-f013:**
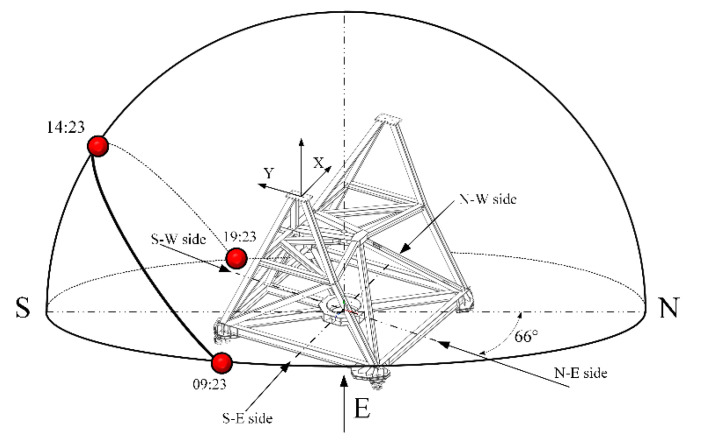
The antenna alidade’s orientation and the apparent motion of the sun on 4 February 2019.

**Figure 14 micromachines-14-01283-f014:**
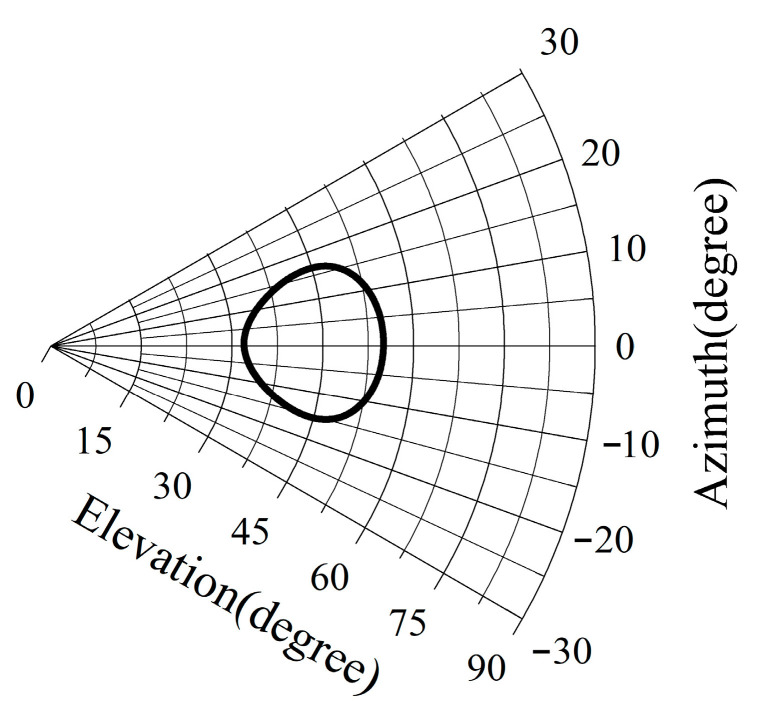
AZ and EL trajectory of Pulsar 1800 + 7828.

**Figure 15 micromachines-14-01283-f015:**
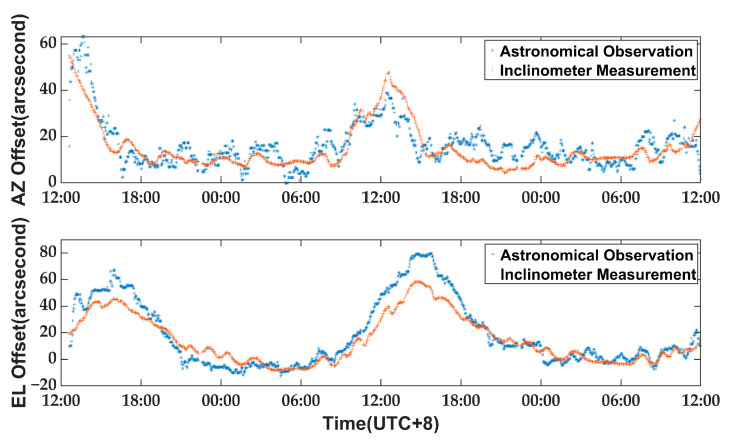
A comparison of thermally induced pointing errors measured by astronomical observation and by the inclinometer measurement system: pointing errors in AZ and EL direction.

**Figure 16 micromachines-14-01283-f016:**
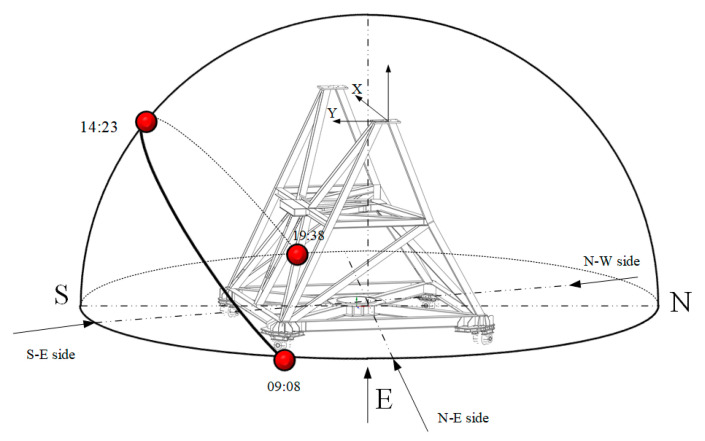
The antenna alidade’s orientation and the apparent motion of the sun on 14 February 2019.

**Figure 17 micromachines-14-01283-f017:**
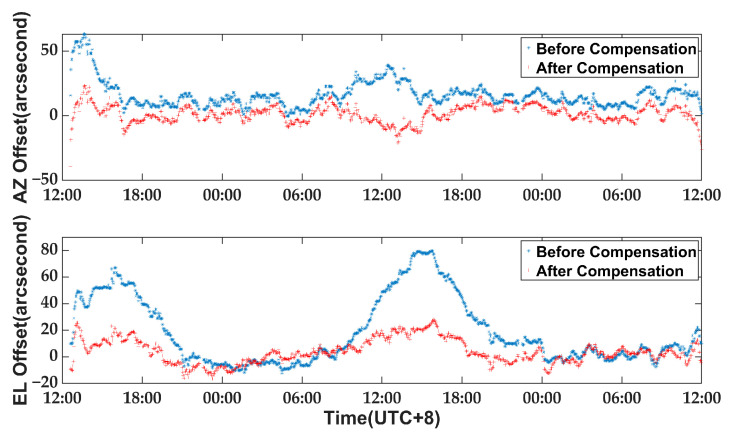
The comparison of pointing errors between these two astronomical observations: Pointing errors in AZ and EL direction.

**Table 1 micromachines-14-01283-t001:** The technical data of Leica Nivel210.

	Nivel 210
Measuring range	−1.51 to +1.51 mrad
Resolution	0.001 mrad
Zero-point stability	0.00471 mrad
Working temperature	−20 to +50 °C

**Table 2 micromachines-14-01283-t002:** The RMS values of the three-night inclinometer measurements.

	X Direction	Y Direction
First night	0.51″	0.56″
Second night	0.83″	1.32″
Third night	0.33″	0.52″

**Table 3 micromachines-14-01283-t003:** The RMS value of the antenna pointing errors in EL direction and AZ direction.

	EL Direction	AZ Direction
First round	11.5″	12.0″
Second round	10.9″	11.4″
Third round	11.7″	12.0″
Average	11.3″	11.8″

**Table 4 micromachines-14-01283-t004:** The RMS value of pointing errors before and after compensation.

	EL Direction	AZ Direction	Total
Full time before compensation	31.4″	19.3″	36.8″
Full time after compensation	9.8″	6.7″	11.9″
Before compensation (14:00–17:00)	64.2″	22.3″	67.9″
After compensation (14:00–17:00)	17.1″	7.7″	18.7″

## Data Availability

Not applicable.

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
