# Peer review of "Measurement and Correction of Pointing Error Caused by Radio Telescope Alidade Deformation based on Biaxial Inclination Sensor"

_micromachines, 2023, doi:10.3390/mi14071283_

Round 1
Reviewer 1 Report
The authors proposed an article on the topic of "Measurement and Correction of Pointing Error caused by Radio Telescope Alidade Deformation based on Biaxial Inclination Sensor" that shows an excellent achievement in reducing the pointing error caused by alidade. There are some comments that should be fixed.
1. No latest references reported, please cite the latest articles from 2023 and 2022 years.
2. Define LEICA, and Figure 5 and its caption must be on the same page.
3. The authors discussed the thermal deformation in the month of February, with 53.4 arcseconds, how much change will occur in the month of July (summer)? can the authors please elaborate?
4. How the relative error (epsilon) becomes |1-cos(beta)|?
Reviewer 2 Report
Review on Measurement and Correction of Pointing Error caused by Radio Telescope Alidade Deformation based on Biaxial Inclination Sensor
I have completed my review of manuscript, entitled, “Measurement and Correction of Pointing Error caused by Radio Telescope Alidade Deformation based on Biaxial Inclination Sensor.”
The degradation of antenna pointing accuracy in radio telescopes can be attributed to several factors, with the deformation and tilt of antenna alidades being particularly significant. This deformation and tilt primarily occur due to the unevenness of the tracking system and the presence of thermal gradients. The accurate pointing of a radio telescope is essential for obtaining precise astronomical observations and data. However, various issues can hinder the performance of the antenna, leading to a reduction in pointing accuracy. Among these issues, the deformation and tilt of the antenna alidades play a crucial role. This paper introduces a system that utilizes a biaxial inclinometer and an environment control box to measure pointing errors. The system enables the precise measurement of alidade tilt, which can be attributed to both track unevenness and thermal gradients.
The subject and findings of this article are interesting and useful. The figures are understandable and the results and well described. Before making a positive decision, I have some concerns and comments about the present form of the manuscript that must be addressed first to improve the quality of the manuscript.
Comments for authors
Comment 1: When reading the abstract, the novelty of this work may not be readily apparent. Therefore, I strongly urge the authors to explicitly state the unique contributions and innovations of their research.
Comment 2: The literature review in this manuscript is limited, with only 18 references being provided to cover the entire content. Furthermore, a significant portion of these references appears to be outdated. To enhance the quality of the literature review, it is strongly recommended to expand it within the introduction section by incorporating citations from recent articles. Expanding the literature review will enable the authors to present a comprehensive and up-to-date overview of the relevant research and developments in the field.
Comment 3: For improved comprehension, it is advisable to merge Figures 4 and 5 into a single figure.
Comment 4: How does the measurement accuracy of the inclinometer system change with slight deviations from the preset installation angle? Describe properly
Comment 5: What are the specific improvements achieved by implementing the environment control box in terms of reducing system noise and enhancing zero-point stability?
Comment 6: How does the accuracy of the inclinometer system compare to other existing methods for measuring pointing errors in radio telescopes? Include in the conclusion section.
Comment 7: Are there any additional applications or potential benefits of the inclinometer measurement system beyond enhancing antenna pointing precision?
Comment 8: The paper contains errors and typos that make it difficult to understand and distort its intended meaning. I encourage authors to reread carefully and fix any grammatical errors.
The paper contains errors and typos that make it difficult to understand and distort its intended meaning. I encourage authors to reread carefully and fix any grammatical errors.
Round 2
Reviewer 2 Report
The authors have addressed my comments and concerns in the revised version. I recommend accepting the paper for publication.